# Impacts of Community Resilience on the Implementation of a Mental Health Promotion Program in Rural Australia

**DOI:** 10.3390/ijerph17062031

**Published:** 2020-03-19

**Authors:** Josephine de Deuge, Ha Hoang, Katherine Kent, Jonathon Mond, Heather Bridgman, Sarah Skromanis, Laura Smith, Stuart Auckland

**Affiliations:** Centre for Rural Health, School of Health Sciences, University of Tasmania, 7250 Launceston, Australia; josephine.dedeuge@utas.edu.au (J.d.D.); Ha.Hoang@utas.edu.au (H.H.); jon.mond@utas.edu.au (J.M.); heather.bridgman@utas.edu.au (H.B.); Sarah.Skromanis@utas.edu.au (S.S.); la.smith@utas.edu.au (L.S.); stuart.auckland@utas.edu.au (S.A.)

**Keywords:** community, evaluation, implementation, mental health, promotion, program, resilience, rural

## Abstract

Mental health promotion programs are important in rural communities but the factors which influence program effectiveness remain unclear. The aim of this mixed-methods study was to assess how community resilience affected the implementation of a mental health promotion program in rural Tasmania, Australia. Four study communities were selected based on population size, rurality, access to local support services, history of suicide within the community, and maturity of the mental health promotion program. Data from self-report questionnaires (n = 245), including items of Communities Advancing Resilience Toolkit (CART) assessment, and qualitative (focus group and interview) data from key local stakeholders (n = 24), were pooled to explore the factors perceived to be influencing program implementation. Survey results indicate the primary community resilience strengths across the four sites were related to the ‘Connection and Caring’ domain. The primary community resilience challenges related to resources. Qualitative findings suggested lack of communication and leadership are key barriers to effective program delivery and identified a need to provide ongoing support for program staff. Assessment of perceived community resilience may be helpful in informing the implementation of mental health promotion programs in rural areas and, in turn, improve the likelihood of their success and sustainability.

## 1. Introduction

Mental health promotion programs are particularly important in rural communities, where the impact of mental health problems are compounded by geographic isolation and a lack of relevant services [1]. Differences in attendance at health services between rural and urban populations may be attributed to an assumed ‘stoicism’ among rural residents, yet their lack of attendance is more accurately attributed to factors such as travel distance, stigma, cultures of self-reliance and the lack of anonymity [2]. Socio-cultural factors [2] such as stigma and/or a reluctance to seek help, are often identified as the major barriers to timely access of support and treatment [3,4,5]. Similarly, standard approaches to health promotion may be less relevant for those in rural and remote areas because of the lowered access to services and community infrastructure that support and promote campaigns. Additionally, individuals living in rural and remote areas may experience a real and/or perceived lack of privacy and confidentiality in matters relating to their health. While mental health programs have been developed specifically for rural communities, the effectiveness of these programs, and the factors likely to influence program effectiveness, remain unclear.

One factor that has been identified as being integral to the success of mental health promotion programs is “community resilience”, a term that refers to the intrinsic capacity of individuals and communities “to stretch or flex in response to the pressures and strains in life encompassing normative stress from daily life to adverse events” [6]. Community resilience has been characterized by four interrelated domains namely: (i) connection and caring; (ii) resources; (iii) transformative potential; and (iv) disaster management [7]. Community resilience is a measure of a community’s response to change with a view to reinstating, maintaining, or enhancing community wellbeing [8,9,10,11]. A high level of community resilience is characterized by the existence of mechanism(s) which lessen the impact of adverse events and elicit a relatively positive psycho-social outcome, despite such events [12]. Major contributing factors to the resilience of a rural community include robust social networks and support, learning experiences, positive environment and lifestyle factors, existing infrastructure and support services, a sense of purpose, and strong leadership [9].

Tasmania is an island state of Australia, lying south of the mainland. It has a population of approximately 520,000 and consists of primarily rural and semi-rural regions. It has the second highest age-standardised rate of suicide in Australia [13] (17.0 per 100,000 people, compared with a national average of 11.7 per 100,000) with more suicide cases reported in rural areas [14]. Hence, mental health promotion is a key concern for local and state governments and non-government organisations. In 2016, a non-profit organization which helped individuals, families and the community through mental health issues with a focus on suicide prevention, established a mental health promotion program “Healthy and Resilience Communities (HaRC)” (referred to throughout as ‘the program’). The program aimed to enhance mental health and wellbeing community protective factors, such as coping capabilities, resilience and connectedness in rural Tasmanian communities. The program involved the establishment of nine local Community Reference Groups (CRGs) in rural communities across Tasmania. CRGs were community-led structures with the purpose of facilitating the development and delivery of health, wellbeing and suicide prevention activities in each community. In 2017, a research team from the Centre for Rural Health, University of Tasmania were commissioned to undertake an evaluation of the program [15]. This paper outlines the methods used to conduct this evaluation and reports on the key findings arising from it. The aim of the evaluation was to assess how community resilience impacted on the program implementation in four program sites in rural Tasmania, and to explore the barriers and enablers related to the successful implementation of the program.

## 2. Materials and Methods

The study employed a realist evaluation framework and a mixed-methods approach (including a quantitative survey, interviews and focus groups) in four communities in Tasmania (more detail on these methodologies is provided below). Realist evaluation focuses on what works, in which circumstances and contexts, and for whom [16]. It allows decision makers to assess whether interventions that prove successful in one setting may be so in another setting, and assists program planners in adapting interventions to suit specific contexts [17]. A mixed-methods approach was employed to provide for a richer understanding of individuals’ views than would be provided by either quantitative or qualitative data collected in isolation [18], and is highly recommended for use in determining the effectiveness of suicide prevention programs [19,20,21].

### 2.1. Study Sites and Participants

At the time of the evaluation the program was established to varying degrees in nine Tasmanian regional communities. Four of the nine communities were selected as case study sites for the evaluation (herein referred to as sites “A”, “B”, “C” and “D”). Sites were selected based on a set of criteria developed in consultation with the funding body who wanted diversity with regards to maturity of the program; population size; geographical location or rurality; access to local mental health support services; history of suicide within the community, existence and maturity of CRGs. The maturity of the program and CRGs was determined by the level of engagement with the program (e.g., the number of forums or interactions the CRG have had with the community) and the period of duration that the program had been established within the community. The study sites varied by population size (n = 485 to 4347). All communities were classified as 4 or “outer regional” on the Australian Standard Geographical Classification-Remoteness Area (ASGC-RA), which classifies locations in terms of remoteness [22]. Three of the target sites, A, B and C, had established local CRGs in place. Sites A and B had been established for a longer period of time, with CRGs having developed terms of reference and meeting more regularly. The fourth site, D, was at a relatively earlier level of development and was in the stage of establishing a CRG.

The study participants included residents of each of the communities aged 18 and above, key stakeholder groups including local and outreach mental health service providers, local council representatives, sporting groups and representatives from local neighbour house establishments. 

### 2.2. Quantitative Data Collection 

The study involved administration of a quantitative survey to a convenience sample of local community members. Convenience sampling was used to collect survey data from community members at each site. As a nonrandom sampling method, this approach allowed easy access to participants who were local community members recruited based on the geographical criteria of being a resident in one of the sites [23,24]. Under this approach, a stall promoting the survey was set up at areas of interest in the community including at shopping centers and pharmacies, where members of the community were invited to fill out paper based surveys on the spot or access an online version via a web-link address. 

The primary measure employed in the survey component of the research was the Communities Advancing Resilience Toolkit (CART) [25]. The CART survey is a publicly available, theory-based and evidence-informed approach to community intervention designed to enhance community resilience by bringing stakeholders together to address community issues in a process that includes assessment, feedback, planning, and action [25]. These principles align with those embedded in the realist evaluative framework [26]. For the current study, the 24-item version of the CART survey instrument [27] was employed as a means of assessing key aspects of perceived community resilience at each of the four evaluation sites. The CART survey measures perceived resilience in five domains: connection and caring; resources; transformative potential; disaster management; and information and communication. Five response options allowed respondents to indicate agreement with each survey item along a range “strongly disagree” to “strongly agree”, which was coded into respondents who agreed or strongly agreed (or otherwise) with each item on the CART survey. To reflect the Australian context in the current study, minor changes to the wording of certain items were made. The survey also queried demographics (age, sex, overall health rating, relationship status, education, employment details, country of birth, language, and whether participants were of Aboriginal or Torres Strait Islander descent). Several additional items were included in addition to the survey tool (questions 25–29) to capture perceptions of overall community resilience and were developed by the authors for the current study. The additional questions focused on perceived barriers and enablers to building resilience such as need for help to become more resilient, experience of isolation, and the perceived importance of isolation and mental health problems within the target communities. These data were coded into respondents who agreed or strongly agreed (or otherwise), reported very often or always (or otherwise), or very or extremely (or otherwise) depending on the question. Data were analyzed using IBM SPSS Statistics for Windows, Version 24.0 (IBM Corp. Armonk, NY, USA), The data analyses were largely descriptive. Chi-square tests were used to assess differences in the proportions of survey respondents across the four study sites who “agreed” with the CART item with respect to levels of perceived community resilience in each of the domains assessed. Post-hoc tests were employed to identify the source of any significant group differences. A statistical significance (alpha) level of 0.05 was employed for all tests and all tests were two-tailed.

### 2.3. Qualitative Data Collection and Analysis

Both semi-structured interviews and focus groups with selected participants were conducted at each site determined by participant availability, accessibility, and preference. Both methods were considered due to pragmatism about recruitment in light of reluctancy within some small communities to participate in qualitative research [28,29]. Participants for focus groups were recruited from third parties who could be key informants on local community, who had local knowledge of the program and/or were key stakeholders, e.g., local mental health support services and snowball sampling. Interview participants were recruited representatives from local neighborhood houses, the program staff, and local council. Participants received an email invitation, together with an information sheet containing details about the study and contact details of the team members. An interview guide (Table 1) was developed based on the relevant literature [30] and aims of the study. 

Interviews and focus groups were audio-recorded and transcribed verbatim into Microsoft Word and then crosschecked against audio recordings. Each participant was assigned an ID number to maintain confidentiality. The data were then imported in QSR-NVivo v10.0 software [31] and analyzed using the six phases of thematic analysis [32]. Transcripts were read and re-read to search for meaning and patterns and notes were taken for coding ideas. All data were then initially coded and collated. The different codes were sorted into potential themes and all the relevant coded data extracts within the identified themes collated. The themes were further reviewed and refined to identify their relationships. The themes were then defined and named. The analysis was completed by two members of the research team. Data were then considered ready for interpretation. The results were compared and discussed at regular meetings among the full research team until agreement was reached. The consolidated criteria for reporting qualitative research (COREQ) were used as a guide for reporting [33].

Data collection activities were coordinated between August and October 2017. The Human Research Ethics Committee (Tasmania) Network (H0016676) granted approval for the study.

## 3. Results

### 3.1. Quantitative Results

Surveys were received from 268 individuals aged 18 to 93 years old. Data for 10 participants, who had unacceptably high levels of missing data (defined as >/= 10% missing data on one or more key study variables), were excluded. An additional 13 participants filled out surveys despite not residing in any of the study sites and were consequently excluded. This was a result of the convenience sampling methods used, as these participants were likely to have been visitors in the community on the day of data collection. The demographic characteristics of the respondents are shown in Table 2.

Table 3 shows the percentage agreement with Communities Advancing Resilience Toolkit (CART) items by study site. Across the four sites, the highest percentage of agreement (80.5% agreement), and therefore the primary community resilience strength, was associated with survey item 4: “People in my community help each other”. Other primary community resilience strength was associated with survey item 1: “People in my community feel like they belong to the community” (70.2% agreement).

The lowest percentage of agreement (18.2% agreement), and thus the primary community resilience challenge, was associated with survey item 7: “People in my community trust public officials.” Other community resilience challenges were associated with survey items 7: “My community has the resources it needs to take care of community problems” (24.7% agreement) and 8: “My community has effective leaders” (25.7% agreement).

Significant differences between sites were observed on several items of the CART. For items related to community resources: item 7 (“my community has the resources it needs to take care of community problems”) and 9 (“people in my community are able to get the support services they need”), participants from sites C and D reported lower levels of agreement (i.e., lower levels of perceived community resilience) than those in site A and B (*p* = 0.01 and *p* = 0.02). Participants from site A also reported higher levels of agreement (i.e., higher levels of perceived community resilience) than those in each of other study sites for item 11 which evaluates community connections (“my community works with organizations and agencies outside the community to get things done”) (*p* = 0.01). There were no differences between sites in percentage agreement on any of the additional survey items included, i.e., items assessing perceptions of overall community resilience, isolation and mental health issues (items 25–29).

### 3.2. Qualitative Results

Individual interviews and focus groups were conducted with 24 participants (3 focus groups and 10 interviews) across the four sites. The focus group participant numbers and gender split is highlighted in Table 4. In addition, eight face to face interviews were conducted with external stakeholders.

Two main themes were derived from thematic analysis and were either enablers or barriers to the implementation of the program. The qualitative results are presented below under the subheadings “enablers” and “barriers”.

#### 3.2.1. Enablers

Three sub-themes were identified under the Enablers theme: Leadership, Flexibility and Community Mobilization.

##### Leadership

Participants identified stable leadership and community development as two critical skills for the local program facilitator, with the personal nature of the program facilitator role being identified as a particularly important enabling factor. Participants understood that program staff changes were sometimes inevitable but expressed the importance of having appropriate handovers and communication during these periods of change:

*“She [the program facilitator] is very passionate about suicide. And she’s a good community development person. So, she came in with all guns charging or whatever and did it very well and she’s a good motivator and all of that……. So, I wonder when the leader leaves, is it worth revisiting where this goes now?”* (Focus group, Participant 4, Site C).

##### Flexibility

Flexibility of access to services was a key strength of the program. Community members were able to access the services offered through the program from their own homes through on-site visits from program staff:

*“One of the things am a big supporter of is people being able to access help in their home and there are not many agencies who will do that, so I think that is a real benefit. The flexibility is great.”* (Focus group participant 1, Site A).

##### Community Mobilization

The program was regarded by some participants to be a catalyst for mobilizing the community to stage public events aimed at raising awareness of suicide prevention and mental health in general. 

*“[community event] showed that we can all come together with what has happened and try really hard to look out for people and make sure that it doesn’t happen again or we can minimise it happening again.”* (Focus group, participant 1, Site B). 

The presence of the program and the associated support structures (such as CRGs) was considered valuable in not only creating conversations in the community about mental health and suicide prevention but this also helped to bring community members in the staging of wellness activities: 

*“It was really amazing to get so many people from different parts of the community to be in one room and listen and accept what it was that we were talking about. That it [suicide/mental health] doesn’t just affect one part of society it crosses over all parts of society.” *(Focus group, participant 1, Site B).

It was noted that these conversations often involved demographic groups that were hard to engage.

*“Men don’t really talk about things as much as women do so I think to have an AFL player there was good. It was almost justification well if it’s ok for him to talk about it ok for us.”* (Interview, participant 7, Site A).

Focus group participants felt that their local CRG required further support in the form of facilitating future activities for the program to remain viable and effective in the future. 

*“Not everyone has the ability to help, I would love to help with something like that but I’m time poor…”* (Focus group, participant 6, Site D).

#### 3.2.2. Barriers

Several sub-themes were identified under the Barriers theme and included: High Staff Turnover, Lack of Ongoing Support, Community Committee Fatigue, Administration Burden, Reduced Relevance within Communities, Program structure and Lack of Community Champion/s.

##### High Staff Turnover

High turnover of program staff had a negative impact on building relationships and trust. Regular interactions between the program staff and the local community were regarded as a key factor contributing to the success of the program. Focus group participants from three out of four study sites identified changes to the program personnel as an issue. Changes in program staff negatively impacted on the community as it required rebuilding of relationships and trust between the facilitator and committee: 

*“The organization changes as well in their managers and setups and so forth, coordinators, and that impacted on the group, I think. Because you only just get a relationship with that coordinator and then they’d be gone for some reason, and it was like rebuilding, several times.” *(Interview, participant 3, site C).

##### Lack of Ongoing Support

Focus group participants at one of the sites highlighted the need for more support from the program staff, especially following a change of facilitators. One of the focus groups identified the potential benefits from the provision of further training in “all aspects of mental health” for the CRG committee members:

*“I feel that the [program] group in this community was left to flounder after the first facilitator left and we were offered no support even though we were one of the pilot programs and we were recognized as a community at risk.” *(Focus group, participant 1, site A).

*“More training for the committee members around all aspects of mental health, so how you approach it with kids, dealing with dementia, having a board knowledge of all types of mental health not just suicide prevention.”* (Focus group, participant 2, site B).

##### Community Committee Fatigue

Participants cited “fatigue” and maintaining motivation and involvement amongst community members as a challenge to implementing the Program. Focus group participants who were members of the program committees observed that the number of people who came to the program meetings had dropped since the first meeting. 

*“At first, we had some huge meeting with lots of men folk involved but now it’s diminished to a small group like this (6–7 people). I think despite our best intentions it’s been really hard to get together. …I just don’t know how you implement it in a community like this where it’s the same people who get asked to do the same things and none of us have the time to do it.”* (Focus group, participant 5, site B).

Participants discussed feeling overwhelmed when dealing with the idea of suicide and the repercussions for the community. Participants discussed this as a potential contributing factor to the membership decrease of the program group:

*“…we don’t like the crisis response part of the program. We don’t feel we are qualified, educated, skilled or should be considering all of the service organisations out there that are here to respond.”* (Focus group, participant 4, site A).

##### Administration Burden

Administrative processes associated with maintaining local CRGs were considered to be taxing and cumbersome. Focus group members representing one of the CRG from one of the sites expressed feelings of being overburdened with paperwork particularly during the set-up stage of the program, which hindered their ability to organize activities:

*“We got, unfortunately, very bogged-down with the actual literature, how it’s going to work in terms of what our terms are, what our conditions are.” *(Focus group, participant 5, site C).

##### Reduced Relevance within Communities

Difficulties associated with sustaining community interest and drive was cited as a barrier to implementing the program over the longer term: 

*“There may be a complacency now that hasn’t been, well, a couple of suicides but they’re very low key, older, not quite so prominent.”* (Focus group, participant 5, site C).

Some focus group participants believed that their community took a very reactive, as opposed to a proactive, approach to suicide prevention in their community. This made it difficult to maintain levels of interest and create a driving force within the community for initiatives such as this program. Though there had been several suicides in the communities previously, participants from one community stated the community had ‘moved on’, while participants from another community discussed suicide not being a prominent issue within their community: 

*“…the [program] model that was being adopted was coming out of a community that had had quite a few suicides that had sparked the program. … But our community didn’t and doesn’t have that driving underlying force of social, there’s not that being driven from the community because we didn’t have a spate of suicides.” *(Interview, Participant 5, site D).

##### Program Structure

Lack of planning around sustaining the groups was cited as a barrier with concerns about the fly-in, fly out (FIFO) approach to service provision. There were concerns around the inconsistency of external services, with the perception that services came into the communities and were then withdrawn. 

*“…it’s a trust relationship that has to be established and because they feel like they’ve been let down in the past, they’re less and less likely to engage and they’re suspicious of engaging, which then means that they either don’t engage or engage with a minimal sense and that service has to withdraw.”* (Interview, participant 4, site D).

This affected the community perception of services in a negative way. This suggests that services within the community need to be made sustainable over the long term to develop trust from the community

*“Whatever we do…I’m really mindful to try and make it very sustainable, so we don’t take big steps, we take smaller sustainable consistent steps.”* (Interview, participant 1, site A).Lack of Community Champion/s

Participants commented that community leaders are also essential to drive community events and activities that the program committee are trying to organize:

*“Groups in this town only work if there’s a driver. And we haven’t got a driver. And the driver doesn’t necessarily have to be seen as a community leader either, it’s just someone that does get stuff out to people.” *(Interview, participant 4, site C).

The qualitative data provided a rich narrative of community experiences in four rural areas associated with the implementation of a rural mental health support service. The data highlighted the challenges and opportunities of engaging and sustaining community involvement in programs that have a primary aim of enhancing coping capability, resilience and connectedness, to better equip rural communities to react to challenging life experiences.

## 4. Discussion

This study employed a mixed-methods approach to examine community resilience affecting the implementation of a mental health promotion program in four rural Tasmanian communities. The survey results indicated the primary community resilience strengths across the four sites were related to the connection and caring domain, with most survey respondents agreeing that people in their communities feel like they help each other and that they have a sense of belonging. These survey results were supported by qualitative data, particularly the community mobilization sub-theme that emphasized the positive benefits of communities facilitating mental health promotion events and generating conversations about suicide.

Conversely, the CART survey results showed primary community resilience challenges were related to resources, with overall most respondents (~75%) perceiving that their community does not have the resources it needs to take care of community problems. There were significant differences in the responses from participants from sites C and D than those in site A and B, who reported lower levels of agreement (indicating lower levels of perceived community resilience) related to resources. With relation to the program, it is important to note that sites A and B have well established and active CRGs in comparison to sites C and D. Resource challenges were congruent with the qualitative data, where sub-themes such as High Staff Turnover, Lack of Ongoing Support, and Community Committee Fatigue were viewed as key program barriers.

Both survey and interview findings suggested that leadership was a factor that affected the implementation of the program. Of note, the CART survey results showed that overall only 25% of respondents agreed that their community had effective leaders, and this was not different across the sites. The existence of strong and effective program leadership structures within the community is frequently cited in the literature as a key factor influencing community readiness [34] and community resilience [35]. Effective leadership at the community level encourages the development and growth of the program. Local leadership structures also provide a conduit through which other services and program may link with the program. Building links with respected leadership structures within communities help programs establish credibility and provide leverage for further development within the community. While connecting with leadership structures is important for promoting social capital and can contribute to community resilience, it is important to recognize that this cannot build resilience in isolation [35], and a reliance on existing leadership structures in communities is associated with danger of over-commitment and burn-out of those in leadership roles [36].

The perceived flexibility of the program delivery was cited as an enabling factor supporting the establishment of the program. Similarly, successful mental health programs have reported that flexibility in service delivery is a major consideration in rural circumstances specifically, by allowing program facilitators the ability to respond to the uniqueness and the similarities of each situation. Ideally, programs should be developed and supported to provide rural clients with a high degree of flexibility to utilize the services they consider useful, and when [37]. Conversely, other features of the program structure, including the perceived ‘FIFO approach’ to service provision were cited as barriers to the program implementation. Previous research has highlighted the need for mental health services in rural areas adopt a multi-pronged, community-based approach to service provision incorporating trusted ‘frontline’ agencies and services, who are supported by accessible secondary level health and welfare services. At the same time, communities must be realistic about the range of services that can be expected, where community size is critical to balance the population needs with the sustainability of services.

The availability of infrastructure and support services have been shown to be a necessary component of community resilience [38]. The community in which program implementation was most advanced also had higher levels of community connections and collaboration, in comparison to the less established sites. Understanding a person’s attachment to—their sense of—community is important to the social and support networks individuals create in communities, and can facilitate recovery from the effects of adverse events [39]. Community structures offer support and identity derived from those nearby or with whom there are meaningful ongoing interactions. A sense of community includes components such as membership, feelings of emotional safety with a sense of belonging and identification; influence, exertion of one’s influence on the community with reciprocal influence of the community on oneself; integration and fulfillment of needs [38]. Social epidemiologists have demonstrated how community connections, belonging, networks, cohesion, and social capital play a significant role in the health, well-being, and mental health outcomes of populations and sub-groups [39] where community participation and sense of community, are positively associated with mental health outcomes.

Another finding from this study was the need to ensure on-going support for program staff [40], with this program experiencing a high staff turnover. Staff shortages and high rates of turnover could be due to a number of reasons including lack of adequate peer support [41] and the recruitment of staff who either do not possess adequate skills [42,43,44] or are perceived by the community as not possessing adequate skills for their roles [45]. Programs that demonstrate success in improving mental health outcomes, by contrast, have highly qualified staff, and provide opportunities for staff to gain additional training in topics specific to the program [46]. Similarly, the program CRG members reported committee fatigue and were burdened with administration. Effective governance, management and leadership have been consistently identified as priorities for successful implementation of health care programs [47], and governing committees contribute significantly to service sustainability. Therefore, adequate support of the CRGs after their establishment is imperative for the program in the longer-term. 

Our findings suggest that community champions are important to mental health promotion program success in our Tasmanian communities, and a lack of identified champions was a major barrier to the program implementation. Kilpatrick and Wilson [48] have shown that ‘boundary crossing champions’ (i.e., those who have local credibility across two or more public, private or community sectors) are respected, trusted, and valued within their communities. Such boundary crossing champions are therefore important source of social capital and could be important as appropriate mental health supports within their communities during difficult times. This is an example of how targeted planning of formal mental health support services in rural communities requires an understanding of the informal interactional infrastructures that are unique within each community, so that social capital is captured to achieve desirable outcomes of community mental health programs, and this should be a consideration when planning future mental health promotion programs.

Kenny et al. highlighted the importance of consultative and transparent processes in engaging communities [49]. Findings from this study suggest that there are mixed views about the degree of consultation and transparency around the program and many respondents revealed a distinct lack of understanding about the program and its operations and achievements. This is not surprising as it has been identified in other literature [50] that local residents are more likely to rely on and trust local sources of information such as outreach workers than sources that are distant and unfamiliar to them. Qualitative findings suggested the lack of, and/or poorly targeted, information and communication as a key barrier to effective delivery of the program. This highlighted the need to better promote the program within the target communities and develop a communication and marketing strategy that provides clear and consistent messages about the program. Availability and accessibility of information and communication systems support dissemination of information within the community and contribute to the knowledge bank within the community. 

The strengths of the current research relate the use of the realistic evaluative approach and mixed methods approach, which contributes to the current literature by highlighting the strengths and opportunities for improvement of a rural mental health promotion program. Importantly, knowledge of community resilience in relation to the program should lead to positive outcomes for communities and individuals in rural Tasmania. Limitations of the current study need to be considered when interpreting the findings. In particular, survey participants were recruited by means of convenience sampling and, due to the study timeline sample sizes were relatively small. The generalizability of the survey findings to the total populations of the communities sampled is therefore unclear. Further, assessment was conducted at only one-time point. Lastly, our recruitment methods may not have been fully inclusive, for example the use of email invitation for participants for focus groups and interviews may have excluded those without email access.

## 5. Conclusions

Our study showed that assessment of perceived community resilience, including features such as community connection and caring, and resources may be helpful in informing the implementation of mental health promotion programs in rural areas and, in turn, improving the likelihood of their success and sustainability. Further research is needed to identify which features of community resilience are most important within each community and how best to assess it in rural settings. Future investment and research is also needed to understand the longer-term outcomes of rural, locally tailored mental health programs to understand the impact at a population level.

## Figures and Tables

**Table 1 ijerph-17-02031-t001:** Interview and focus group interview guide.

Questions
What does a strong, robust and responsive community look like?
Can you provide examples of events or experiences that have impacted negatively on the collective health and wellbeing of the community?
Which groups (e.g., aged people 65+, people with disabilities, LGBTI groups, farmers) in the community do you believe may be at particular “risk” to mental health issues and why?
How do you believe the community and individuals can best respond to minimize the impact of these events? (e.g., resources, networks, relationships, projects, groups, organizations)
What resources and assets were particularly helpful or unhelpful?
What was learnt that could be useful in the future?
Given these learnings how can the community better prepare, respond and empower itself against the impacts of traumatic events or occurrences into the future?
What do you know about the work of Rural Alive and Well and in particular the Healthy and Resilient Communities program?
Does your community have a community suicide prevention and wellbeing management program? If so, what can you tell me about it?
What are the key ingredients to an effective community suicide prevention and wellbeing management program?
Are you aware of the Healthy and Resilient Communities program in your community? (list their activities)
If yes, have you or your community been involved in any of the program’s activities?
How does a program such as [name of the program] best invest its resources (people, funds, time) into your community too build the collective capacity of the community to better prepare and respond to life changing events? (e.g., professional development, advice, resources, events, funding or support for sharing practice)
Partnerships are one of the key elements of the program’s model, what could a partnership model look like in your community?
How could the success or otherwise of the program be measured?

**Table 2 ijerph-17-02031-t002:** Demographic characteristics of survey participants (n = 245) by evaluation site.

	Evaluation Site							
	A (n = 54)	B (n = 69)	C (n = 84)	D (n = 38)		Total		
**Demographic Characteristics**	**Mean (SD)**	**Mean (SD)**	**Mean (SD)**	**Mean (SD)**		**Mean (SD)**	***F***	***p***
**Age (years)**	47.5 (16.1)	51.9 (15.4)	52.0 (14.5)	49.1 (17.4)		**50.51 (15.62)**	**0.88**	**0.48**
	**%**	**%**	**%**	**%**		**%**	**χ^2^**	***p***
**Sex**								
**Male**	14.8	50.7	38.1	28.9		**35.1**	**26.9**	**<0.01**
**Female**	85.2	49.3	61.9	71.1		**64.9**		
**Health**								
**Poor**	7.4	1.5	6.0	0.0		**4.1**	**14.23**	**0.58**
**Fair**	20.4	17.6	13.3	18.4		**16.9**		
**Good**	33.3	41.2	30.1	31.6		**34.2**		
**Very good**	29.6	27.9	36.1	28.9		**31.3**		
**Excellent**	9.3	11.8	14.5	21.1		**13.6**		
**Relationship Status**								
**Single, never married**	7.4	11.8	10.8	13.2		**10.7**	**12.24**	**0.43**
**Married or living as married**	81.5	61.8	66.3	52.6		**66.3**		
**Single, separated or divorced**	3.7	16.2	13.3	23.7		**13.6**		
**Other**	7.4	10.3	9.6	10.5		**9.5**		
**Main Activity**								
**Paid work full-time**	29.6	27.5	23.8	34.2		**27.8**	**32.19**	**0.12**
**Paid work part-time**	14.8	26.1	21.4	13.2		**20.0**		
**Full-time student**	5.6	1.4	0.0	0.0		**1.6**		
**Home duties/caring for children**	14.8	5.8	8.3	13.2		**9.8**		
**Retired**	25.9	21.7	29.8	23.7		**25.7**		
**Seeking paid work**	1.9	5.8	2.4	2.6		**3.3**		
**Other**	7.4	11.6	14.3	13.2		**11.8**		
**Involved in Agriculture/Farming**								
**Yes**	35.2	20.6	22.0	2.6		**21.5**	**14.46**	**0.01**
**No**	64.8	79.4	78.0	97.4		**78.5**		
**Highest Level of Education**								
**Year 10**	31.4	15.2	19.5	22.9		**21.4**	**22.31**	**0.32**
**Year 12**	13.7	15.2	17.1	8.6		**14.5**		
**Trade certificate/apprenticeship**	17.6	24.2	25.6	25.7		**23.5**		
**Undergraduate diploma**	17.6	12.1	14.6	20.0		**15.4**		
**Bachelor’s degree**	5.9	15.2	14.6	8.6		**12.0**		
**Postgraduate degree or diploma**	13.7	18.2	8.5	14.3		**13.2**		
**Country of Birth**								
**Australia**		88.7	88.1	82.9	84.2	**85.8**	**3.88**	**0.42**
**Other**		11.3	11.9	17.1	15.8	**14.2**		
**First Language**								
**English**		100.0	100.0	97.6	100.0	**99.2**	**18.99**	**<0.01**
**Other**		0.0	0.0	2.4	0.0	**0.8**		
**Aboriginal or Torres Strait Islander Descent**								
**Neither**		88.5	77.9	88.5	91.9	**86.0**	**15.50**	**0.49**
**Aboriginal**		7.7	17.6	9.0	5.4	**10.6**		
**Torres Strait Islander**		0.0	0.0	1.3	0.0	**0.4**		
**Aboriginal and Torres Strait Islander**		0.0	1.5	1.3	2.7	**1.3**		
**Prefer not to say**		3.8	2.9	0.0	0.0	**1.7**		

**Table 3 ijerph-17-02031-t003:** Percentage (%) agreement with Communities Advancing Resilience Toolkit (CART) items by study site.

		Evaluation Sites						
		A (n = 54)	B (n = 38)	C (n = 69)	D (n = 84)	Total	χ^2^	*p*
Age Mean (SD)		47.5 (16.1)	49.1 (17.4)	51.9 (15.4)	52.0 (14.5)	**50.51 (15.62)**		
Sex %	Male	14.8	28.9	50.7	38.1	**35.1**		
	Female	85.2	71.1	49.3	61.9	**64.9**		
	**Connection and Caring**							
1	People in my community feel like they belong to the community	63.5	71.1	73.8	71.2	**70.2**	**1.59**	**0.66**
2	People in my community are committed to the well-being of the community	54.9	52.6	64.6	64.4	**60.4**	**2.57**	**0.46**
3	People in my community have hope about the future	57.7	50.0	56.3	61.1	**57.1**	**1.28**	**0.73**
4	People in my community help each other	78.0	81.6	79.4	82.9	**80.5**	**0.53**	**0.91**
5	My community treats people fairly no matter what their background is	56.0	39.5	35.4	37.5	**41.3**	**5.88**	**0.12**
**Resources**								
6	My community supports programs for children and families	58.0	73.7	72.3	68.5	**68.1**	**3.43**	**0.33**
7	My community has the resources it needs to take care of community problems (for example, money, information, technology, tools, raw materials, and services)	34.6	39.5	18.8	15.1	**24.7**	**12.08**	**0.01**
8	My community has effective leaders	35.3	23.7	18.5	26.4	**25.7**	**4.35**	**0.23**
9	People in my community are able to get the services they need	51.0	44.7	33.8	25.0	**36.7**	**10.00**	**0.02**
10	People in my community know where to go to get things done	37.3	44.7	32.3	30.6	**35.0**	**2.53**	**0.47**
**Transformative Potential**								
11	My community works with organizations and agencies outside the community to get things done	54.9	36.8	33.8	26.0	**36.6**	**11.10**	**0.01**
12	People in my community communicate with leaders who can help improve the community	46.9	50.0	30.8	37.5	**39.7**	**5.07**	**0.17**
13	People in my community work together to improve the community	49.0	63.2	63.1	61.6	**59.5**	**3.02**	**0.39**
14	My community looks at its successes and failures so it can learn from the past	35.3	36.8	26.6	33.3	**32.4**	**1.56**	**0.67**
15	My community develops skills and finds resources to solve its problems and reach its goals	35.3	39.5	26.2	31.9	**32.4**	**2.13**	**0.55**
16	My community has priorities and sets goals for the future	35.3	39.5	27.7	36.6	**34.2**	**1.90**	**0.59**
**Disaster Management**								
17	My community tries to prevent disasters	39.2	51.4	41.3	55.6	**47.1**	**4.47**	**0.22**
18	My community actively prepares for future disasters	32.0	36.8	26.2	47.2	**36.0**	**7.03**	**0.07**
19	My community can provide emergency services during a disaster	60.8	70.3	56.3	69.4	**63.8**	**3.45**	**0.33**
20	My community has services and programs to help people after a disaster	43.1	55.5	29.7	47.2	**42.7**	**7.49**	**0.06**
**Information and Communication**								
21	My community keeps people informed (for example, via television, radio, newspaper, internet, phone, neighbours) about issues that are relevant to them	52.9	47.4	67.7	49.3	**55.1**	**6.17**	**0.10**
22	If a disaster occurs, my community provides information about what to do	45.1	44.7	42.9	47.2	**45.1**	**0.26**	**0.97**
23	I get information/communication through my community to help with my home and work life	38.8	47.4	46.9	27.4	**38.8**	**6.93**	**0.07**
24	People in my community trust public officials	25.5	15.8	17.2	15.3	**18.2**	**2.42**	**0.49**
**Additional Items**								
25	My community is resilient ^i^	54.2	73.7	70.3	63.9	**65.3**	**4.58**	**0.21**
26	My community needs help to become more resilient ^i^	56.3	55.3	53.1	57.1	**55.5**	**0.23**	**0.97**
27	How often do you experience isolation? ^ii^	16.7	18.4	10.1	11.9	**13.5**	**2.10**	**0.55**
28	How big a problem do you think isolation is in your community? ^iii^	42.6	26.3	37.7	29.8	**34.3**	**3.84**	**0.28**
29	How big a problem do you think mental health issues, such as anxiety and depression are in your community? ^iii^	48.1	60.5	50.7	47.6	**50.6**	**1.93**	**0.59**

^i^ Percentage of participants who responded to the item statement with “agree” or “strongly agree”; ^ii^ Percentage of participants who responded to the item statement with “very often” or “always”; ^iii^ Percentage of participants who responded to the item statement with “very” or “extremely”.

**Table 4 ijerph-17-02031-t004:** Focus group participant numbers and gender.

Site	Number of Participants	Male n (%)	Female n (%)
A	7	0	7 (100)
B	6	1 (17)	5 (83)
C	4	2 (50)	2 (50)
D	7	5 (71)	2 (29)
Total	24	8 (33)	16 (66)

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
