# Peer review of "Impacts of Community Resilience on the Implementation of a Mental Health Promotion Program in Rural Australia"

_ijerph, 2020, doi:10.3390/ijerph17062031_

Round 1
Reviewer 1 Report
Introduction
First page line 33 the authors put in brackets with e.g. examples of structural factors but do something different for socio-cultural factors; I would suggest doing this consistently
Line 36 remain rather than remains unclear
Line 41-2 use of capitals; reconsider.
The authors emphasise the need for mental health promotion in rural communities where incidence of suicide are higher (Intro line 53). However, there is earlier acknowledgement that services in rural communities are inadequate to address need and there is no engagement with literature documenting that actually, the issue is not that people in these communities don’t seek help, they do and when they do their needs are not addressed. So health promotion may encourage people to seek help, but that is not useful if there is no help. I would suggest the authors engage with this to strengthen and better position their work.
Line 63 and reports on rather than to report. Also here, was the aim to assess whether the program increased or improved community resilience (and if so, based on what? How determined?), ‘related factors’ is too vague and explore rather than exploring, and finally were you also interested in what hindered as well as what facilitated? Much more clarity and specificity is needed around the aim and purpose of the evaluation, and the paper.
Materials and methods
There is a lot of ambiguity and missing detail in this section. The term ‘survey form’ is confusing. You mention the four sites but it is only later down that you explain this. It is very unclear who the focus group and interview participants were, and why focus groups and individual interviews were chosen – these methods produce different kinds of data usually for different reasons so you need to explain the rationale used here. Perhaps and for whom sentence ending in line 71. Finally here greater explanation/justification is needed for the last sentence in this section.
Study sites
You say the selection of the sites was made based on a predetermined criteria, who made this criteria and why, what was the rationale? E.g. you say population size was a criteria but later down you say there were varying population sizes. Much more explanation is needed around these criteria and the selection process so your readers understand your thinking.
When you say lower level of development, do you mean earlier on in the development? What do you mean by convenience sampling in the next line? Give more detail/explanation so we understand the process.
Quant data collection, line 93 you say the 24 item version but it is not until later that this information is contextualised/explained. Perhaps flip this/change the order of information so the process is clearer.
Qual data collection and analysis
You say e.g. local mental health support services, give more detail here, like who else? Based on what rationale? The process of using an email invitation only raises equity issues that are not discussed. This automatically excludes though without email. This needs to be acknowledged, or if other avenues were available, this needs to be detailed.
Much more detail is needed for how analysis was done. What happened to the focus group data? How was coding and categorising actually done?
Quant results
What is meant by unacceptably high? Be more specific. How did you end up with participants who weren’t living in the communities?
Qual results
This section needs a fair amount of work. It appears like results have just been copied and pasted in from a report. It is not sufficient to just insert quote after quote, you need to do some analytical work in the presentation of the data, lead your reads in, and make a narrative.
You need to be specific about how many participants were in each focus group and one focus group is one unit of analysis. I would strongly encourage the authors to be more critical about the phrasing ‘themes that emerged’ – consider this statement in relation to the below quote from Braun and Clarke’s article entitled ‘Reflecting on reflexive thematic analysis’:
Themes do not passively emerge from either data or coding; they are not ‘in’ the data, waiting to be identified and retrieved by the researcher. Themes are creative and interpretive stories about the data, produced at the intersection of the researcher’s theoretical assumptions, their analytic resources and skill, and the data themselves.
This may help you in re-working this section overall.
Discussion
Line 252 services has shown is confusing phrasing. Later down more advanced, again I would think about what you are really meaning here, the longer they have established?
It is interesting that you begin to talk about understanding a person’s attachment to or sense of community – there is a substantial body of literature on these concepts that would be useful to engage with, but there is really no engagement at this point.
Line 264 as a factor confusing phrasing again.
Line 272 You say ‘The literature’, what literature is this? Only one citation.
Much more thought needs to be given to the ‘limitations’ paragraph, especially in relation to the qualitative data. In general, this section needs development. Interpretations and reflections need to be deepened with greater engagement with existing literature and how this work contributes to the existing body of knowledge.
Conclusion
Related factors again too vague, especially for this part of a manuscript, and generally, too vague.
Reviewer 2 Report
This has the potential to be a valuable publication. However, in its current form it lacks a clear thread and fails to capitalise on the value that mixed methods can offer. The qualitative findings need reworking and the discussion needs to bring together the qual and quant results. I look forward to reading the next iteration.
Line 64: It's not clear from the aim how community resilience relates to 'the program' being discussed within the context of this paper (e.g. are you looking at how the program has affected community resilience? are you looking at community resilience independently of the program? why was this chosen as the the primary survey measure? etc.)
Line 100: Missing basic details about the CART e.g. Likert scale? Bivariate response? This only became clear once I read the legend in Table 3.
Line 112: Needs detail about what determined individual interview or focus group (quite different methods which may elicit different responses)
Line 131: How was 'unacceptably' high levels of missing data determined?
Qualitative results - the qualitative evidence has not been well analysed. The themes are too broad, and the interpretation and selection of quotes as evidence is weak. Some line-by-line feedback is provided below.
Line 171-73: Needs italics to identify this as a quote. The statements here also require an additional/alternative quote demonstrating the effect on trust, etc. Not demonstrated by the existing quote.
Line 197-99: Require a quote as evidence of this statement
Line 207: Remove italics
Line 197-206: This doesn't speak to the them 'barriers'. May be better understood as evidence of 'program relevance' or 'particular community needs/responses'
Line 213-15: Example quote does not support the statement
Line 216-221: The authors have identified the opportunities for stronger linkages, while the research participants identified the problem of poor communication. This needs rephrasing and authors interpretation of the need for improved communication should then come in the discussion
Line 223-24: Statement needs evidence.
Line 230-234: Evidence in quote does not match preceding statement i.e. participants evidence suggests that a driver is essential but that this can be other than the community leader. This would seem an important distinction in this context and open up the possibility of other community members being effective in driving change.
Line 235-36: Sentence doesn't make sense. Also isn't clear whether this contains a quote??
Line 237-238: Not a sentence.
Line 222-241 all rely on evidence from Participant 14. Is there evidence from other participants to support this?
Line 264-65: Sentence doesn't make sense.
Discussion: There is currently nothing in the discussion that draws the quantitative and qualitative evidence together. This is an important factor of mixed methods research where drawing on these two methods provides a richer picture, rather than two separate views. e.g. the discussion could draw together conclusions about the relationship between reduced community resilience related to resources (quantitative results) and the limited availability of trusted services (qualitative results)
Round 2
Reviewer 2 Report
Some significant improvements have been made to this paper, notably in better linkages between quantitative and qualitative findings. Still some work to go as outlined below.
Line 36 - missing 'be'. Also a very long and confusing/repetitive sentence.
Line 82-85 - quantitative data here did not allow for conclusions about causality or generalisability. Not sure of the purpose of including this?
Line 111 - 'were' should be 'was'
Line 119 - The statement 'These principles align with those embedded in the realist evaluative framework.' should be referenced
Line 138 - Should this read 'data analyses were largely descriptive'?
Line 145-47 - The statement 'in light of reluctancy within some small
communities to participate in qualitative research' should be referenced
Table 2 and 3 - I find Table 2 and 3 difficult to interpret. Is there a clearer way to represent the Chi Square and significance measures? Could you shade/bold the 'total mean', Chi Square and significance columns to distinguish these from the individual site columns?
Line 190 - missing 'the'?
Table 3 - item 7 missing 'the'
Line 225-26 - need a quote to support this claim
Qualitative research reporting requires quotes as evidence of the claims being made. This is still not consistently provided throughout the results section. Please review this throughout.
In addition, quotes need to included at the relevant sections, rather than at the end of a long paragraph (e.g. community mobilization - the first quote is good evidence of community events bringing people together, but it comes in the text after a sentence on hard to engage demographic groups.....poorly placed). Same problem in 'Lack of ongoing support'. 'Community Committee Fatigue' manages this better.
While the authors claim that a range of participants voices are heard via the quotes, there is no evidence for this provided. There needs to be some coding used after each quote to show that the quotes are from different participants/focus groups.
Line 243 - quote does not speak about flexibility
Still a few typos identified.
Author Response
Please see attached letter
